# Inertial displacement of a domain wall excited by ultra-short circularly polarized laser pulses

T. Janda[1,2], P.E. Roy[3], R.M. Otxoa[3], Z. Šobáň[2], A. Ramsay[3], A.C. Irvine[4], F. Trojanek[1], M. Surýnek[1], R.P. Campion[5], B.L. Gallagher[5], P. Němec[1], T. Jungwirth[2,5] & J. Wunderlich[2,3]

Domain wall motion driven by ultra-short laser pulses is a pre-requisite for envisaged low-power spintronics combining storage of information in magnetoelectronic devices with high speed and long distance transmission of information encoded in circularly polarized light. Here we demonstrate the conversion of the circular polarization of incident femtosecond laser pulses into inertial displacement of a domain wall in a ferromagnetic semiconductor. In our study, we combine electrical measurements and magneto-optical imaging of the domain wall displacement with micromagnetic simulations. The optical spin-transfer torque acts over a picosecond recombination time of the spin-polarized photo-carriers that only leads to a deformation of the initial domain wall structure. We show that subsequent depinning and micrometre-distance displacement without an applied magnetic field or any other external stimuli can only occur due to the inertia of the domain wall.

[1] Faculty of Mathematics and Physics, Charles University, Ke Karlovu 3, 121 16 Prague 2, Czech Republic. [2] Institute of Physics, Academy of Sciences of the Czech Republic, Cukrovarnická 10, 162 00 Praha 6, Czech Republic. [3] Hitachi Cambridge Laboratory, J. J. Thomson Avenue, Cambridge CB3 0HE, UK. [4] Microelectronics Research Center, Cavendish Laboratory, University of Cambridge, Cambridge CB3 0HE, UK. [5] School of Physics and Astronomy, University of Nottingham, Nottingham NG7 2RD, UK. Correspondence and requests for materials should be addressed to J.W. (email: jw526@cam.ac.uk).

**D**omain walls (DWs) driven by short field[1] or current[2,3] pulses of length $\sim$1–10 ns and moving at characteristic velocities reaching $\sim$0.1–1 µm ns$^{-1}$ (ref. 4) are displaced over the duration of the pulse by distances at least comparable but typically safely exceeding the domain wall width. In this regime inertia, causing a delayed response with respect to the driving field and a transient displacement after the pulse, is not the necessary pre-requisite for the device operation and is rather viewed as negative factor. It can set the operation frequency limit of the DW device and potentially affect precise positioning of the DW by the driving pulse. Realizing massless DW dynamics is therefore one of the goals in the research of field-driven and current-driven DWs[5].

The aim of our study is the demonstration of a micrometre-scale DW displacement by circularly polarized, ultra-short laser pulses (LPs). Our experiments are in the regime where the external force generated by the LP acts on the picosecond timescale over which the expected sub-nanometre DW displace-ment would be orders of magnitude smaller than the DW width and insufficient for any practical DW device implementation. Inertia allowing for a free transient DW motion is the key here that enables the operation of the DW devices in the regime of the ultra-short optical excitations, rather than being a factor limiting the operation of the opto-spintronic DW devices.

Our study links the physics of inertial DW motion with the field of optical recording of magnetic media. The manipulation of magnetism by circularly polarized light, demonstrated already in ferrimagnets[6], transition metal ferromagnets[7] and ferromagnetic semiconductors[8], has become an extensively explored alternative to magnetic field or current-induced magnetization switching. Our work demonstrates that optical recording can in principle be feasible at low power when realized via an energy-efficient DW displacement driven by ultra-short LPs and without the need to heat the system close to the Curie temperature.

The III–V-based ferromagnetic semiconductor used in our study is an ideal model system for the proof of concept demonstration, as well as, for the detailed theoretical analysis of the DW dynamics in this new regime. DWs in out-of-plane magnetized (Ga,Mn)(As,P) have a simple Bloch wall structure with low extrinsic pinning[9]. The non-thermal optical

spin-transfer torque (oSTT) mechanism, which couples the circular polarization of the incident light to the magnetization via spin-polarized photo-carriers is microscopically well understood in this ferromagnetic semiconductor material[10]. In our experiments, individual circularly polarized $\sim$100 fs short LPs at normal incidence and separated by $\sim$10 ns expose an area with a single DW. As illustrated in Fig. 1a, the generated perpendicular-to-plane spin-polarized photo-electrons exert the oSTT only in the region with an in-plane component of the magnetization, that is, in the DW. The action of the oSTT is limited by the photoelectron recombination time $\sim$10 ps.

To probe the inertial DW motion, we make use of elastic properties of a uniformly propagating DW. In this case, the DW propagates continuously and remains connected so that the local DW pinning affects the entire wall over its whole extension. First, the Oersted field generated in a stripe line above the magnetic bar nucleates a reversed magnetic domain. Then, a single DW is driven towards a cross structure by a small external magnetic field of a slightly larger magnitude than the propagation field $B_{PR}$. The low $B_{PR}$ of $\sim$0.1 mT found in our bar devices patterned from an epitaxially grown $Ga_{0.94}Mn_{0.06}As_{0.91}P_{0.09}$ 25 nm thick film implies a very small DW pinning on structural defects and inhomogeneities. In this case, DW propagation is uniform and a straight DW becomes pinned at the entrance of the cross structure as shown in Fig. 1b. To continue the DW propagation through the cross, the DW must increase its length which is accompanied by an increase in its magnetic energy. This results in a restoring force which can be expressed in terms of a virtual restoring field $B_R(x)$ that depends on the position $x$ of the DW. Here, $B_R(x)$ acts as to always drive the DW back to the cross entrance. The magnetic field-driven expansion of a DW pinned at the cross entrance is analogous to the inflation of a two-dimensional soap-bubble (Fig. 2a). The DW depins when the applied field exceeds the maximum restoring field $B_R^{max}$ (ref. 11). Within this model, $B_R(x)$ reaches its maximum value $|B_R^{max}| = \sigma/(M_S \cdot w)$ at the cross centre at $x = 0$ (Fig. 3a) and the DW can only depin once it passes the cross centre. Here, $\sigma = 4\sqrt{AK_E}$ is the DW energy per unit area, $K_E$ the effective perpendicular anisotropy coefficient, $A$ the exchange stiffness, $M_S$ the saturation magnetization and $w$ is the width of the bar. The DW can be depinned from the cross by either an applied magnetic field $B_A > |B_R^{max}|$ or by the oSTT. We can therefore use $|B_A| \leq B_R^{max}$ to calibrate the strength of the oSTT. First, however,

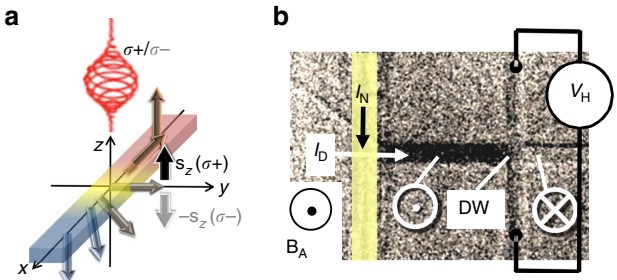

**Figure 1 | Optical spin-transfer torque on a magnetic domain wall.** (**a**) Sketch of light helicity-dependent optical spin-transfer torque on a DW. Optically generated spin-polarized photo-electrons exert spin-transfer torque only on the rotating magnetization of the DW in the perpendicular magnetized film. Outside the DW, electron spin-polarization and magnetization are collinear. (**b**) Differential MOKE image of the initialized DW position where the DW is geometrically pinned at the cross entrance of a 4 µm wide Hall bar. After saturation, a reversed domain is nucleated by the Oersted field generated by the nucleation current $I_N$. Subsequently, the single DW propagates to its initial position when applying a small magnetic field of $B_A \sim$ 0.2 mT. The initial straight DW position can also be detected by a AHE measurement when applying the current $ID$ along the Hall bar. The corresponding Hall signal $V_H$ corresponds to $\sim$11% of the total signal on compete magnetization reversal.

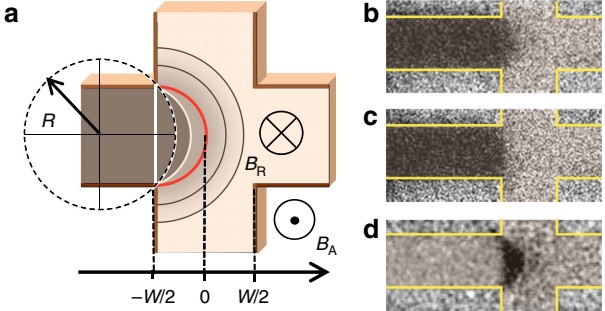

**Figure 2 | Soap-bubble-like domain wall expansion.** (**a**) Schematic sketch of soap-bubble-like extension of an elastic DW within a symmetrical cross under the application of a magnetic field: The domain wall stays pinned on the input corners until it reaches the cross centre (red half-circle) (Supplementary Note 1). (**b**) Differential MOKE images of domain configurations in case of a geometrically pinned DW in a 6 µm wide device at $B_A = 0.25$ mT (**b**) and after the field has been switched off ($B_A = 0$ mT) (**c**). (**d**) Bubble-like domain shape when subtracting **c** from **b**. Note that the DW switched back after the field has been switched off.

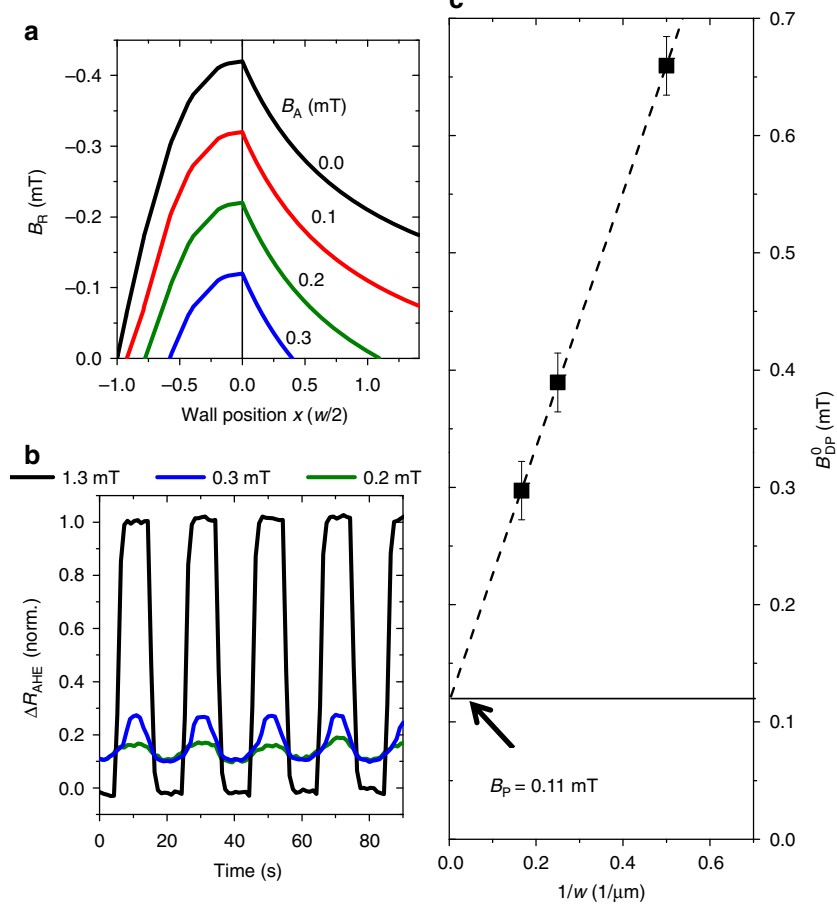

**Figure 3 | Repulsing motion of a geometrically pinned domain wall.** (**a**) Effective restoring field $B_R$ as a function of DW position opposing wall propagation at various applied magnetic fields $B_A$ calculated for a $w = 4\,\mu m$ wide cross bar device. $|B_R|$ becomes maximal when DW reaches the cross centre. (**b**) Relative change of AHE signal (normalized to the total AHE signal on compete magnetization reversal) of a $4\,\mu m$ wide device due to elastic DW repulsive motion driven by an alternating field excitation $B_A = B_0 |\sin(\omega \cdot t)|$ with $B_0 = 0.2\,mT$, (green), $B_0 = 0.3\,mT$, (blue). AHE signal for complete magnetization reversal with $B_A = 1.3\,mT \cdot \sin(\omega \cdot t)$, (black). (**c**) Experimentally determined depinning fields of three different devices with bar widths of $w = 2$, 4 and $6\,\mu m$. $B_{DP}^0$ corresponds to the lowest applied magnetic field necessary to depin the DW from the cross without LP irradiation and is equal to $B_R^{max} + B_{PR}$, $B_{PR}$ is the DW propagation field of the unpatterned magnetic film. The broken line is a linear fit of the experimentally obtained data points and corresponds well to the theoretical prediction from our simple propagation model (Supplementary Note 1). The error bars of the depinning fields correspond to the s.d. derived from 10 individual measurements.

we have to confirm the elastic nature of DWs in our devices, and verify the applicability of the bubble-like DW model of Fig. 2a.

## Results

**Elastic domain wall pinning.** We performed magnetic field-driven DW motion experiments without optical excitation. Depinning fields for three different devices with bar widths of 2, 4 and $6\,\mu m$ are shown in Fig. 3c as a function of the inverse bar width. The slope of the linear fit agrees with that obtained from the measured effective perpendicular anisotropy, $K_E = 1{,}200\,J\,m^{-3}$, the saturation magnetization, $M_S = 18\,kA\,m^{-1}$, and assuming the exchange stiffness, $A = 50\,fJ\,m^{-1}$, which is a reasonable estimate for our GaMnAsP film[9]. The elastic behaviour of the $\pi\sqrt{A/K_E} \sim 20\,nm$ wide DW is also confirmed by MOKE images of the $6\,\mu m$ wide bar device shown in Fig. 2b–d. In Fig. 2b, the DW bends into a bubble-like shape under the influence of an applied field $B_A = 0.25\,mT$. Figure 2c shows that the restoring field drives the DW back to the cross entrance after $B_A$ is turned-off. Fig. 2d displays the difference between the two MOKE images in Fig. 2b,c, confirming the bubble-like shape of the DW. In addition, anomalous Hall effect (AHE) measurements performed on the $4\,\mu m$ device under an alternating field

excitation $B_A = B_0 |\sin(\omega \cdot t)|$ also confirm the elastic DW behaviour (Fig. 3b). If $B_0$ does not exceed $B_R^{max}$, for example, for $B_0 = 0.2\,mT$ (green), and $B_0 = 0.3\,mT$ (blue), the periodic variation of the AHE signal indicates that the DW is at the position $x$ where $B_R(x)$ and $B_A(t)$ compensate. The residual AHE signal at $B_A = 0$ of about 10% of the maximum AHE signal at reversed saturation (DW depinned from the cross) corresponds to the AHE-response for the magnetization distribution with a straight DW located at the cross entrance. (For more details see Supplementary Note 1.).

**Helicity-dependent domain wall excitation by $\sim 100\,fs$ LPs.** We now combine the elastic pinning properties of the DW at the cross with the light-induced excitation experiments to proof the inertial character of the oSTT-induced DW motion. The basic idea of our experiment is to exploit the elastic restoring force that acts continuously throughout the entire cross (of a width up to $6\,\mu m$ in our study) against the expansion of the DW that is driven by individual $\sim 100\,fs$ LPs. The photo-generated electrons can transfer their spin to the magnetization only during their $\sim 10\,ps$ lifetime, which is three orders of magnitude shorter than the pulse separation time of $\sim 10\,ns$. The measurements shown in Fig. 3 are

performed at a 90 K sample temperature. We obtain similar results when performing the measurements also at higher (95 K) and at lower (75 K) sample temperatures, as shown in Supplementary Note 1. At these temperatures, LPs with a wavelength $\lambda = 750$ nm excite photo-electrons slightly above the bottom of the GaAs conduction band so that for a circularly polarized incident light, photo-electrons become spin-polarized with the degree of polarization approaching the maximum theoretical value of 50% (ref. 12). To avoid the difficulty with aligning our $\sim 1\,\mu$m Gaussian spot on top of a $\sim 20$ nm wide DW, we employ the experimental procedure sketched in Fig. 4a. First, a straight DW is positioned at the cross entrance. Then, the LP spot is placed 10 $\mu$m away from the DW on the reversed domain side and a magnetic field $B_A$ with $B_{DP}^0 (\approx +0.4\,\text{mT}) > B_A > B_{PR}(\approx -0.1\,\text{mT})$ is applied. ($B_{DP}^0$ is the DW depinning field without LP irradiation.) In this field range and without LP irradiation, the DW remains pinned at the cross entrance. The LP spot is then swept at a rate of $\sim 2\,\mu$m ms$^{-1}$ for 20$\mu$m along the bar so that the initial DW position is crossed by the spot and $\sim 10,000$ ultra-short LPs time-separated by $\sim 10$ ns expose the DW. The lowest applied magnetic field at which the DW depins from the cross in the presence of LPs is labelled $B_{DP}$. The dependencies of $B_{DP}$ on the LP energy density for circularly polarized $\sigma^+$, $\sigma^-$ and linearly polarized $\sigma^0$ LPs are shown in Fig. 4b. First, we recognize a reduction of $B_{DP}$ with increasing energy density for all three LP polarizations. In case of the linear polarization, that is, without the oSTT contribution, we attribute the reduction of $B_{DP}(\sigma^0)$ only to the LP-induced sample heating. Importantly, we do not observe DW depinning without applying

$B_A > 0$ up to the highest LP energy densities used in our experiments of more than $\sim 30$ mJ cm$^{-2}$. At large LP energy densities above $\sim 20$ mJ cm$^{-2}$, we observe a saturation of $B_{DP}(\sigma^0)$ with increasing LP energy density implying that LP heating does not increase anymore. We assign this behaviour to the saturation of photo-carriers generated at very high LP energy densities.

For circularly polarized LPs, an additional contribution from the oSTT is present. We observe for all measured LP energy densities that $B_{DP}(\sigma^+) < B_{DP}(\sigma^0) < B_{DP}(\sigma^-)$ for the positive magnetization orientation of the nucleated domain. In case of $\sigma^+$-polarized LPs and at high enough LP energy densities (above 12 mJ cm$^{-2}$) the DW depins without an applied magnetic field (and even at small negative applied magnetic field which opposes DW expansion).

For $\sigma^-$-polarized LPs and the same initial domain configuration, we do not observe the zero-field DW depinning up to the highest LP energy density used in our experiments. Instead, we again observe saturation of $B_{DP}(\sigma^-)$ above $\sim 20$ mJ cm$^{-2}$. We attribute the difference in the saturation values of $B_{DP}(\sigma^-)$ and $B_{DP}(\sigma^0)$ to the effect of the oSTT acting against depinning for $\sigma^-$-polarized LPs.

We estimate the LP heating-related temperature increase by comparing $B_{DP}(\sigma^0$, LP energy density, $T = 90$ K) measured at constant 90 K base temperature with the temperature dependence of $B_{DP}^0(T)$ without LP irradiation. We found that for LP energy densities of up to 35 mJ cm$^{-2}$, the temperature increase does not exceed the Curie temperature of $\sim 115$ K of the GaMnAsP film.

The differential MOKE image in Fig. 4c shows an example of the domain configuration after the DW has depinned from the

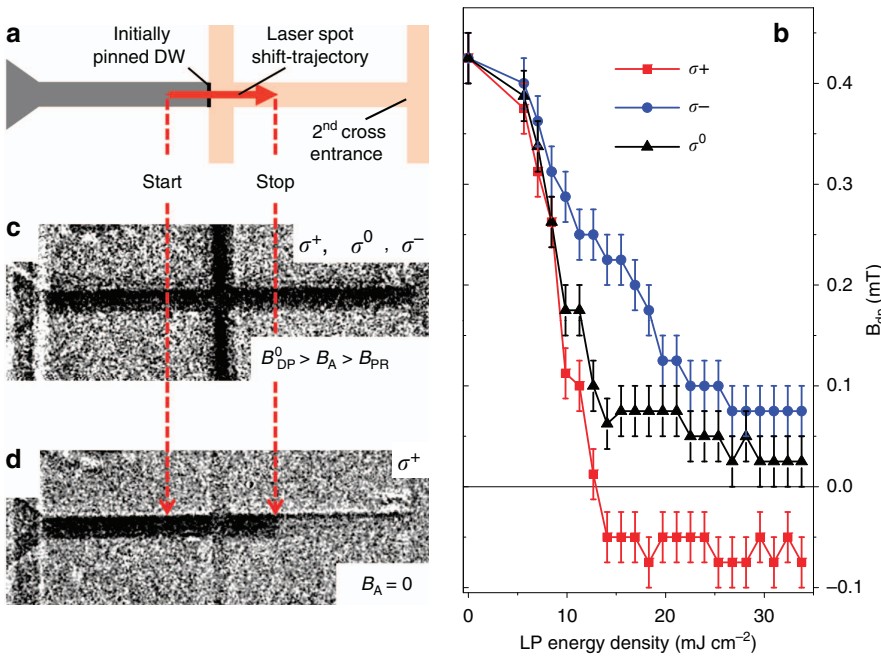

**Figure 4 | Helicity-dependent magnetic domain wall depinning. (a)** Experiment: To obtain $B_{DP}$, we first saturate the magnetization in a strong negative field. Then, a reversed domain is nucleated and a DW is positioned at the cross entrance. The LP spot is now focused to its 'start' position 10 $\mu$m away from the initial DW location within the reversed domain. Subsequently, the spot is swept by 20 $\mu$m along the bar crossing the initial DW position with a velocity of $\sim 2\,\mu$m ms$^{-1}$. Starting from a small negative applied field of $B_A = -0.1$ mT, DW depinning is inferred from AHE measurements and differential MOKE micrographs taken after the laser spot sweep at constant $B_A$. If the DW is still located at the cross entrance, $B_A$ is increased by $+0.025$ mT followed by another laser spot sweep and subsequent AHE and MOKE detection. This procedure is repeated with stepwise increased $B_A$ until DW depinning is detected. Each individual data-point of $B_{DP}$ is obtained as the average from five independent depinning field measurements. The error bars correspond to the maximal observed scatter of $B_{DP}$ around the corresponding mean values. **(b)** Depinning field $B_{DP}$ as a function of LP energy density for circularly left (red), linearly (black) and circularly right (blue) polarized light up to the highest LP energy density where the temperature increase due to LP heating does not exceed the Curie temperature of the magnetic film. Depinning at zero applied field is only observed if the oSTT is generated by $\sigma^+$-polarized LPs. **(c)** Final domain configuration after laser sweeps with an applied field larger than propagation field $B_{PR}$ and **(d)** at zero or small negative applied magnetic field.

cross entrance by optical excitation in conjunction with a constant applied magnetic field $B_A$, which is larger than the DW propagation field of the bar outside the cross. After depinning from the cross irradiated by polarized LPs, the DW becomes pinned again at a second cross which was not irradiated during the experiment. Figure 4d shows the final domain configuration after DW depinning by $\sigma^+$-polarized LPs at zero applied magnetic field. In this case, the $\sigma^+$-polarized LPs depin and drive the DW forward to the final irradiated spot position.

**Inertial domain wall propagation.** From the measurements shown in Fig. 4, we can conclude that for the given initial domain configuration, the oSTT generated by $\sigma^+$ ($\sigma^-$)-polarized LPs assists (opposes) DW depinning. The measurements confirm that only $\sigma^+$-polarized LPs can move the DW beyond the maximum of the pinning barrier at the cross centre. Considering the $\sim 100$ fs short and $\sim 10$ ns time-seperated LPs, depinning of the DW by the oSTT becomes only possible if the elastically pinned DW propagates forward in between successive LPs. Depinning by a DW motion without inertia would require DW velocities of more than $1 \, \mathrm{km \, s^{-1}}$ that are unrealistically high for the DW motion in GaMnAsP films, where the maximum magnon velocity is of similar magnitude, and where we have observed and calculated Walker break down velocities $\sim 10 \, \mathrm{m \, s^{-1}}$ for the oSTT, current and field-driven DW motion[9]. (See also Supplementary Note 2.)

To verify our interpretation, we repeated our measurements at the inverted magnetization configuration in which the reversed magnetization of the nucleated domain points in the negative $\left(-m_z^R\right)$ direction. In this case, the oSTT should act in the opposite direction. Indeed, we observe the opposite helicity dependency in our experiments. Figure 5 shows measurements on a 4 µm wide device comparing the two magnetization configurations. The consistency found between $B_{DP}(\sigma^{+(-)}, +m_z^R) \approx -B_{DP}(\sigma^{-(+)}, -m_z^R)$ and $B_{DP}(\sigma^0, +m_z^R) \approx -B_{DP}(\sigma^0, -m_z^R)$ confirms the oSTT mechanism and the high reproducibility of our measurements.

Note, that a heat gradient can in principle also drive the DW motion[13]. The heat gradient-driven motion can become helicity

dependent if the light absorption in the two adjacent magnetic domains is helicity dependent due to the magnetic circular dichroism (MCD). In our experiments, such a scenario is unlikely because about $\sim 98\%$ of the LP light penetrates through the 25 nm thick magnetic GaMnAsP film and is absorbed and transformed into the heat in the GaAs substrate with no dependence on the helicity.

An indication that the MCD is not the origin of the observed helicity-dependent DW depinning is given by helicity-dependent DW experiments shown in the Supplementary Note 3. The experiments are performed at photon energies ranging from below the band-gap up to high energies where the net spin-polarization of photo-electrons is reduced due to the excitation from the spin-orbit split-off band. We do not observe the helicity-dependent DW depinning at photon energies, where MCD of GaMnAsP is still present while simultaneously the photoelectron polarization is strongly reduced.

To investigate the effect of the MCD on the helicity-dependent DW motion in more detail, we present additional experiments in the Supplementary Note 3, which allow us to identify the sign and estimate the magnitude of the temperature gradient generated by the MCD between two opposite magnetized domains. We found that the MCD-generated heat gradient is smaller than the helicity independent heat gradient generated by the Gaussian LP spot, and more importantly, that the DW motion induced by the MCD is in the opposite direction to the observed helicity-dependent DW motion. This excludes unambiguously the MCD as the origin of our experimental observations.

To further analyse heat gradient-related DW drag effects due to the non-uniform heating by the Gaussian-shaped LP spot, we have performed measurements with opposite laser spot sweep directions. In this case, the heat gradient with respect to the initial DW position is inverted. As shown in the Supplementary Note 3, sweeping the LP spot along the bar from an initial position outside of the nucleated domain to the final position in the nucleated domain does not change the helicity dependency of the depinning field. Additional measurements on devices with 2 and 6 µm wide bars have, apart from the stronger (weaker) DW pinning strength and larger (smaller) temperature increase from LP heating in the 2 µm (6 µm) device, also confirmed that $B_{DP}(\sigma^{+(-)}) < B_{DP}(\sigma^0) < B_{DP}(\sigma^{-(+)})$ for $+(-)m_z^R$.

We now support our interpretation of the experiments by 1-dimensional Landau–Lifshitz–Bloch (LLB) numerical simulations of the magnetization $\mathbf{m}$[14], coupled to the precessional dynamics of the spin-polarized photo-carrier density, $\mathbf{s}$[10]:

$$\frac{\partial \mathbf{m}}{\partial t} = -\gamma \mathbf{m} \times \mathbf{H}_{eff} - \frac{\gamma \alpha_\perp}{m^2} \mathbf{m} \times (\mathbf{m} \times \mathbf{H}_{eff}) + \frac{\gamma \alpha_{||}}{m^2} (\mathbf{m} \cdot \mathbf{H}_{eff}) \mathbf{m} \tag{1}$$

$$\frac{\partial \mathbf{s}}{\partial t} = \frac{-J_{ex}}{\hbar m_{eq}} \mathbf{s} \times \mathbf{m} + R(t)\hat{n} - \frac{\mathbf{s}}{\tau_{rec}} \tag{2}$$

In equation (1), $\mathbf{m} = \mathbf{M}(T)/M_0$, with $M_0$ denoting the saturation magnetization at zero temperature and $\gamma$ is the gyromagnetic ratio. The first, second and third terms describe the precession, transverse relaxation and longitudinal relaxtion of $\mathbf{m}$, respectively. The effective field $\mathbf{H}_{eff} = \mathbf{H}_d + \mathbf{H}_{ex} + \mathbf{H}_{mf} + \mathbf{H}_k + \mathbf{H}_{OSTT} + \mathbf{H}_r$ comprises demagnetizing field, exchange field, internal material field related to longitudinal magnetization relaxation, uniaxial magnetocrystalline anisotropy field, oSTT field and a geometrical pinning field, respectively. The two parameters $\alpha_\perp(T)$ and $\alpha_{||}(T)$ represent the transverse and longitudinal damping, respectively. The oSTT from $\mathbf{s}$ on $\mathbf{m}$ is taken into account by $\mathbf{H}_{OSTT} = \frac{J_{eff}(T)}{\mu_0 m_{eq} M_0} \mathbf{s}$ (with $J_{eff}(T) = J_{ex} m_{eq}^2$). For more details see Supplementary Note 2.

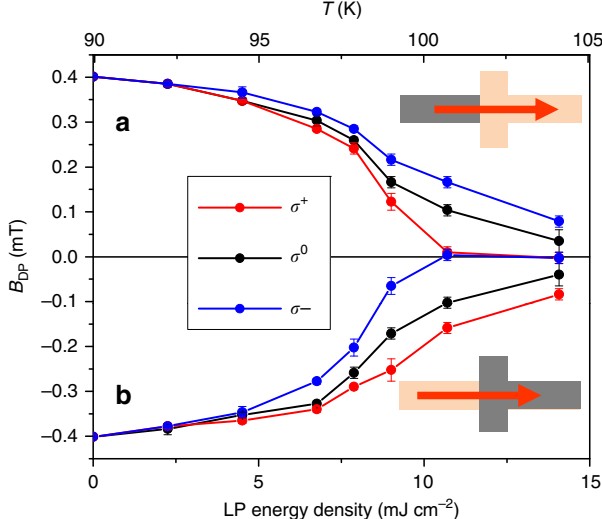

**Figure 5 | Helicity-dependent depinning field at reversed magnetic domain configuration.** Depinning field $B_{DP}$ as a function of LP energy density for circularly left (red), linearly (black) and circularly right (blue) polarized light with positive (**a**) and negative (**b**) nucleated domain magnetization. The LP-related temperature increase estimated from the comparison between $B_{DP}(\sigma^0, T = 90 \, K)$ and $B_{DP}(0, T)$ is plotted at the top of the graph.

Equation (2) describes the time evolution of the spin-polarized photo-electron density $\mathbf{s}$. The first term is the precession of $\mathbf{s}$ around the exchange field of $\mathbf{m}$ with the coupling strength $J_{ex}$; $m_{eq}$ is the equilibrium magnetization normalized by the zero-temperature saturation magnetization $M_0$. The second term describes the spin-polarized photo-electron injection rate $R(t)$, which is non-zero only during the $\sim 100$ fs LP, and $\hat{n}$ is the helicity-dependent spin-polarization of the injected electrons. Depending on the light helicity, $\hat{n}$ is $[00 \pm 1]$. The last term describes the decay of the spin density, determined primarily by the recombination time of the photo-electrons, $\tau_{rec}$.

In the simulations, we consider a Bloch DW subjected to LPs and the restoring field $B_R(x)$ as in Fig. 3a. $B_R^{max}$ was set to a reduced value of 0.1 mT due to heat (deduced from Fig. 4b and described in the Supplementary Information). Figure 6a,b show the simulated time evolution of $\mathbf{m}$ and $\mathbf{s}$ at the initial DW centre during and after the application of a single 150 fs pulse with $\sigma^+$ ($\hat{n} = [0\ 0\ 1]$) polarization.

In Fig. 6a, the fast precession of $\mathbf{s}$ around the exchange field of $\mathbf{m}$ takes place until the photo-electrons recombine. Only during this short time, angular momentum is transferred to $\mathbf{m}$. The precession of $\mathbf{s}$ is much faster than the dynamics of $\mathbf{m}$ so that a significant change of $\mathbf{m}$ due to the precession around $\mathbf{H}_{eff}$ happens after the photo-electrons recombined. Figure 6b shows the time evolution of $\mathbf{m}$ at the centre of the initial DW

($\mathbf{m}$ is initially directed along $+\hat{\mathbf{y}}$ for the Bloch DW). During the short oSTT, $\mathbf{m}$ is only weakly disturbed from its equilibrium direction. It takes $\sim 1$ ns before it is rotated towards the $\hat{\mathbf{z}}$ axis. At this time, the centre of the initial DW becomes part of the reversed domain and the DW has shifted by half width. The deformation of the moving DW from the equilibrium Bloch DW profile is shown in Fig. 6c. The deformation $\Delta \mathbf{m}$ is obtained by subtracting the moving DW from the undisturbed Bloch DW profile after having shifted the centre positions of the two DWs to $x = 0$. Shortly after the LP exposure at $t = 50$ ps, the DW magnetization is strongly distorted. The simulation indicates that even after 5 ns, $\Delta m_x(0) \approx 0.15$, so that the original Bloch DW is still deformed towards a Néel DW. The deformation of the DW from its equilibrium profile long time after the LP was applied causes magnetization precession around the arising effective fields and keeps the DW moving. The photo-electrons only distort the DW, while its subsequent motion is driven by the relaxation of the DW towards its equilibrium profile. In Fig. 6d, the DW position versus time is plotted during the first three LPs. As can be seen, the entire DW moves predominantly between and not during the pulses.

A calculation confirming the depinning of the DW from the cross is shown in Supplementary Note 2. Here, oSTT pulses are applied until the DW reaches the cross centre and overcomes the maximum value of the geometric pinning potential. Our simulations fully confirm the experimental observations and

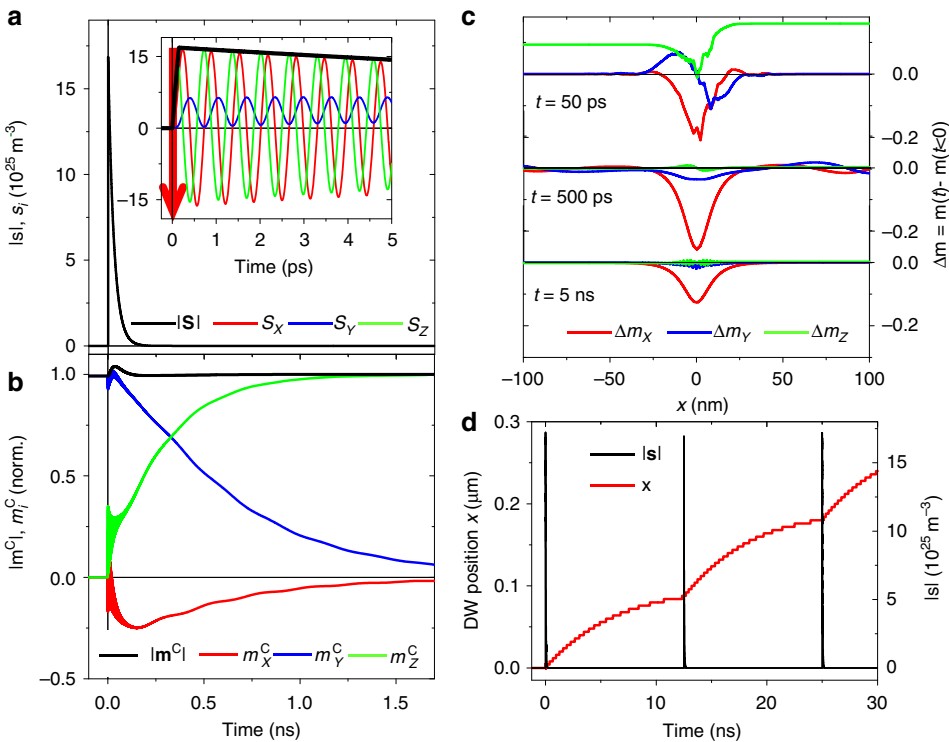

**Figure 6 | Dynamics of coupled photo-electron spin density and DW magnetization.** (**a**) Simulated time evolution of photo-electron spin density $\mathbf{s}$ at the centre of the DW generated by a 150 fs long LP indicated as a red arrow. Inset: the $x$, $y$, $z$ components of $\mathbf{s}$ versus time $t$ showing the fast precession around the exchange field of magnetization $m$. The red arrow in the inset indicates the LP. (**b**) The components of $\mathbf{m}$ and $|\mathbf{m}|$ versus $t$ at a fixed position corresponding to the initial DW centre $\mathbf{m}^C$. At $t = 0$, $\mathbf{m}^C$ is oriented along the $y$ direction at the centre of the Bloch-like DW. Note that $\mathbf{m}^C$ has been normalized by its modulus before the LP is applied. The graph shows a fast initial excitation due to the LP and a damped fast jiggling during the recombination time of the photo-electrons. During this short time, angular momentum is transferred from $\mathbf{s}$ to $\mathbf{m}^C$ causing a deformation of the DW. Note, that during the oSTT, the magnitude of $|\mathbf{m}^C|$ increases due to the interaction between the non-zero $y$ component of the precessing spin density and the magnetization at the DW centre oriented initially also along $y$. (**c**) Time evolution of the DW deformation $\Delta \mathbf{m}$ along the DW width. The three plots show the time evolution of the deviation from the undisturbed DW profile in the rest frame of the domain wall with zero at the DW centre after the pulse was applied. The slowly relaxing DW deformation causes the DW motion. (**d**) The DW position as a function of time for the first three $\sigma^+$-polarized LPs (a pulse occurs every 12.5 ns). In **a–d**, $\tau_{rec} = 30$ ps and $R = 1.2 \times 10^{39}$ m$^{-3}$ s$^{-1}$.

the inferred picture in which the inertial motion is responsible for the DW displacement driven by the ultra-short LPs.

## Discussion

In summary, we have shown photon-helicity-dependent inertial DW motion excited by ultra-short circularly polarized LPs. We found that the domain wall only deforms during the short excitation. After excitation, the DW propagates self-propelled driven by its relaxation back to the unperturbed DW profile. We note that the helicity-dependent DW motion can be also realized by a continuous light excitation as shown in Supplementary Note 3. However, LP-excited inertial DW motion is less affected by pinning and, therefore, more efficient than DW motion excited by continuous and lower energy-density light. This is due to the fast initial acceleration of the heavily deformed DW shortly after the pulse. We also remark that the LP-induced helicity-dependent DW motion is not limited to diluted magnetic semiconductors. The oSTT-induced DW motion may also be realized in heterostructures, where the spin-polarized photo-carrier excitation and spin-transfer torque are spatially separated, for example, when spin-polarized photo-electrons are injected from an optically active semiconductor into an adjacent thin ferromagnetic film. In this case, the oSTT can be equally efficient as found in our present study since the total magnetic moment of a $\sim 1$ nm thin magnetic transition metal film is comparable to the total magnetic moment of our 25 nm thick diluted magnetic semiconductor film with $\sim 5\%$ Mn doping. Indeed, the DW motion in a ferromagnetic film driven by spin-polarized currents applied electrically in the direction perpendicular to the film-plane has been recently proposed[15] and experimentally observed, showing a very fast DW motion[16] and low driving current densities[17]. Our concept represents an optical analogue to these electrical driven DW experiments with the potential of delivering orders of magnitude shorter while still highly efficient spin torque pulses.

## Methods

**Experimental set-up.** We use a wide-field magneto-optical microscope to monitor our magnetic bar devices, and to identify DW nucleation and DW position. The magneto-optical polar Kerr effect measured with linearly polarized light of $\lambda = 525$ nm is used to visualize the magnetization distribution in our bar devices. The $\sim 100$ fs short LPs with 12.5 ns separation time between successive pulses are generated by a Ti:Saphire laser and focused to a $\sim 1$ µm wide spot by an objective lens which is mounted to a 3D piezo-positioner to enable the precise alignment of the laser spot to the magnetic bar device.

**Computational geometry and simulation procedure.** Our simulation is based on the LLB approach[14]. We consider a one-dimensional bar with $4{,}095 \times 1 \times 1$ computational cells composing a structure. The cell dimension is 4 nm $\times$ 4 µm $\times$ 25 nm. A Bloch DW is initialized in the centre of the bar and is let to relax quickly with strong damping by setting a large damping parameter $\lambda = 0.9$ (Supplementary Note 2). This configuration is then used as a starting configuration for the simulations of domain wall motion under the light pulses. Once the domain wall is prepared, circularly polarized light is pulsed at a rate of 80 MHz. The length of each pulse is set to 150 fs. For the simulation of the depinning process, the spin-polarized carrier injection rate is $R = 1.225 \times 10^{39}$ m$^{-3}$ s$^{-1}$. This order of magnitude for $R$ is required for the DW to escape the elastic pinning potential. The equivalent pulse power corresponds to the LP energy density of 5.6 mJ cm$^{-2}$ assuming a skin depth of 1 µm. At this energy density, helicity-dependent DW motion becomes evident in the experiments, cf., Figs 4c and 5. Further, all simulations were done in zero externally applied magnetic field and a damping of $\lambda = 0.01$ was used in all dynamical simulations. Throughout all simulations, a centring procedure is employed that keeps the DW in the middle of the length of the bar. In this way, propagation distances as long as needed can be simulated without having to worry about stray field effects should the domain wall have come close to the edges of the bar or that the DW moves out of the computational region.

**Data availability.** The data sets generated during and/or analysed during the current study are available from the corresponding author on reasonable request.

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

## Acknowledgements

We acknowledge support from European Metrology Research Programme within the Joint Research Project EXL04 (SpinCal); from EU ERC Synergy Grant No. 610115, from the Ministry of Education of the Czech Republic Grant No. LM2015087, from the Grant Agency of the Czech Republic under Grant No. 14-37427G, and by the Grant Agency of Charles University in Prague Grant Nos. 1360313 and SVVĐ2015Đ260216.

## Author contributions

T.Ja., J.W. and M.S. carried out the measurements and P.N., A.R. and F.T. advised on the experimental set-up. P.E.R. and R.M.O. developed the LLB numerical simulations code and performed the micromagnetic simulations. R.P.C. and B.L.G. grown and supplied the GaMnAsP films, Z.Š. manufactured the samples and A.C.I. advised on the sample fabrication. All authors discussed the results and analysed the data. J.W., T.Ju., T.Ja. and P.E.R. wrote the paper. J.W. initiated and coordinated the project.

## Additional information

**Competing interests:** The authors declare no competing financial interests.

**Publisher's note**: 

