## [Peer Review File · Nature Communications]

Reviewer #1 (Remarks to the Author):

The authors report the propagation of magnetic domain walls in GaMsAsP induced by circularly polarised light. The authors use different experimental cases (circular vs linear polarisation, opposite magnetic orientations, different laser sweep directions, different wavelengths) to discriminate the different physical mechanisms at play. From this analysis, the authors convincingly show the effects of "optical spin transfer", i.e. the action on a DW of spin polarised carriers generated with circularly polarised light in a magnetic semiconductor. The main result of this paper is the analysis of how a short light excitation (100fs) can induce large displacements of a DW whose response time is much slower. For this the authors use a LLB model to study the relaxation, 'inertial' movement of a distorted DW in a medium of variable saturation magnetisation.

The article is interesting and easy to follow, and presents a quite complete picture of this complex situation: starting with the demonstration of the excitation mechanism (optical spin transfer) to the dynamics of DWs excited by those mechanisms. My only criticism is that some of the results could be more extensively discussed (as per some of the remarks below).

I recommend the publication of this manuscript after the authors address my minor remarks below. My remarks and questions are roughly in order of appearance in the main text.

1. The 'inertial displacement' of the DW by relaxation from a deformed state has been observed before, and these works should be cited: to my knowledge, it was first shown in J.-Y. Chauleau et al., Phys. Rev. B 82, 214414 (2010).

2. page 4 first paragraph : What do you mean by "DW propagation is coherent"?

3. page 4 last line: in " $K_E = 1200 \text{ Jm}^3$ " should be instead " $K_E = 1200 \text{ J/m}^3$ ".

4. Perhaps state the width of the track in the legend of fig2E.

5. page 5 2nd paragraph: When describing the "basic idea of [the] experiment" you state that the elastic force acts over the $1\mu\text{m}$ wide cross. However, you show measurements shown in crosses of different widths, and indeed the cross described in the paragraph just before is $4\mu\text{m}$ wide.

6. fig 3 (and fig S4): There is little discussion on the evolution of the oSTT effect with LP energy. For example, there's seems to be a saturation of the the effect with LP energy, as seen by the plateau of $(B_{DP+} - B_{DP-})$. Have you got an idea why? Have you perhaps reached the full DW distortion towards a Neel orientation? or perhaps it's due to heating?

7. page 7 para 2: you conclude that sigma- LPs drive the DW away from the cross but, at this paragraph, you have only shown the opposite effect. The results for pulling DWs with sigma- light is shown only later...

8. page 7 para 2: you say that "DW velocities of more $1\mu\text{m/ns}$ [10^3m/s] are unrealistically high". How so? can you estimate the Walker velocity? Is there another limitation? (heating? nucleation?)

9. eq.s 1&2 (page 8-9) : you should write explicitly the coupling term between the two equations (the exchange field term in Heff created by s) in the main text, and not just in the additional materials. It is a central part of this model and critical for the understanding your model.

10. page 9 para 4: from your text, one may think that the action of the photoelectrons on m happens after the recombination ("...so that a significant change of m [...] happens after the photo electrons recombined."). If I understood correctly, the photo electrons only distort the magnetisation of the DW until they recombine, and the DW only moves later (through relaxation)

due to its slower dynamics (?). Perhaps you meant that the major effect of the photoelectrons (the DW propagation) are only evident well after they have recombined?

11. page 10 (& addt mat): you state (and show in the additional materials) that a CW excitation is less power efficient in propagating the DW, something you have seen experimentally and in simulations. A bit more information in the main text as to why is this the case would be interesting.

12. fig 1B: it seems there are some white semi-transparent rectangles over the white domains in the MOKE differential image. This may give the idea to the reader that the experimental image is less noisy than it actually is... I would suggest to replace them by empty shapes with borders, so as to still indicate the position of the structure while still showing your MOKE image.

Reviewer #2 (Remarks to the Author):

The authors report on DW motion induced by circularly polarized light pulses in GaMnAsP bars. This work is a continuation of their previous work [ref 13] published last year. They investigated the mechanism of optical Spin Transfert Torque, where the photon momentum exerts a torque on the DW magnetization leading to DW motion. The novelty of the present work is the use of ultrashort (sub ps) pulses. That offers them access to DW dynamics at the picosecond time scale. In particular they claim to observed an inertial DW motion between the pulses.

This work belongs to the emerging subject of light induced magnetization manipulation, as magnetization reversal or DW motion. A growing number of publications have reported such effects, in both ferromagnets or ferrimagnets. Nevertheless it has emerged recently that heating related effects also play a major role in the magnetization dynamics. In their paper Janda et al., have addressed this issue. However as this point is critical and as the author's arguments to exclude it do not appear convincing enough, I cannot recommend this work for publication in Nature Comms.

My main concern is that all the experiments were performed very close to TC, which makes very hard to exclude any thermal effect as the main source of DW motion. In their previous PRL, the authors selected their substrate temperature to form easily domains. Here they benefited from a strip line for nucleation that could have given them access to lower temperatures. In this paper, unfortunately, no additional measurements were presented at lower temperatures. I would have expected that oSTT to be as efficient, given the very high quality of the sample, in particular very weak pinning energy. That would have given a stronger support to their claim.

At their experimental temperature, the authors did consider heating related effects before rejected them as the main source of DW motion. In particular they dismissed any magnetic circular dichroism related effects based on the fact that they observed DW motion over a spectral range for the pump laser that does not match the MCD one in GaMnAs. I cannot agree with this statement. I could find in the literature several papers that show dichroism curves reproducing the trend in fig S4.b:

For perpendicular magnetization samples, Sun et al. PRB 83, 125206 and Riahi et al. JMMM395,340 measured Kerr angle and Kerr angle + ellipticity respectively vs wavelength. Although Kerr is not MCD, these quantity are highly connected. And they do observe a reduction of the Kerr angle and ellipticity at shorter wavelengths. In another study Dobrowolska et al. Nat. Mat.

11, 444 worked in planar samples with at high perpendicular magnetic fields. They observe a similar trend.

They also rejected this MCD effect based on the fact that most of the light is absorbed into the substrate and therefore the MCD related heating should be weak. However I went through these two publications, Khorsand et al. PRL 108, 127205 and Gorchon et al. arXiv <http://arxiv.org/pdf/1604.06441v1.pdf> . They both assessed the role of MCD in dichroic materials. The latter publication spotted clearly that even a small MCD (0.5 %) could lead to helicity dependent optical magnetic switching driven by heat. In particular they indicate that this effect should be stronger in "materials with large MCD, small dipolar fields, limited in-plane diffusion and strong temperature of the anisotropy close to TC", matching very closely the properties of GaMnAsP. Although they discussed the case of magnetic switching, I believe that a proper analysis of DW motion should be undertaken in the light of this mechanism. In particular, the effect of the laser pulse ellipticity remains tenuous and increase only when approaching TC. The missing temperature dependence experiment does not help to distinguish between heat and oSTT related effects.

Another argument put forward by Janda et al. is the inertial motion observed after the light pulse. Here again, I am a bit concerned as heat dissipation in the magnetic layer and substrate occurs on a time scale far longer than the sub picosecond pulse and the repetition time. That means that right after the pulse, the anisotropy, the pinning energy and the magnetization remain very weak which can lead to very different dynamics under their applied magnetic field, especially high DW velocities.

I was surprised by the negative points in the depinning field difference observed at low power and at 880 and 750 nm in Fig S4A lower plots. Could the authors comment on that?

I found the introduction and justification of this work about the interest of inertial and massless DW motion rather confusing (a massless DW is the best but probably also inertial...).

Reviewer #3 (Remarks to the Author):

In the article "Inertial displacement of a domain wall excited by ultra-short circularly polarized laser pulses", T. Janda et al. describe domain wall motion driven by femtosecond circularly polarized pulse. The driving force was attributed to the optical spin transfer torque. The authors claims that the motion is inertial displacement.

This work is original and interesting, and also important for application. The physical interpretation of the experimental data is mostly convincing. The references are appropriate. The manuscript is well written.

Their experimental finding will stimulate a new trend in the research community. Several future research questions arise; For example, how fast can the domain wall move with this inertial mechanism? How is the inertial mass of the motion defined? What limits the length of the inertial displacement by a laser pulse?

An additional remarks are listed below.

1) The laser heating also lead to inerial displacement of a domain wall. What is the mechanism?

2) In inset in Fig. 3B, the data for linearly polarized LP should be added.

The additional data will provide a strong evidence for oSTT mechanism in the case of circularly polarized LP.

Reviewer #1 (Remarks to the Author):

The authors report the propagation of magnetic domain walls in GaMsAsP induced by circularly polarised light. The authors use different experimental cases (circular vs linear polarisation, opposite magnetic orientations, different laser sweep directions, different wavelengths) to discriminate the different physical mechanisms at play. From this analysis, the authors convincingly show the effects of "optical spin transfer", i.e. the action on a DW of spin polarised carriers generated with circularly polarised light in a magnetic semiconductor. The main result of this paper is the analysis of how a short light excitation (100fs) can induce large displacements of a DW whose response time is much slower. For this the authors use a LLB model to study the relaxation, 'inertial' movement of a distorted DW in a medium of variable saturation magnetisation.

The article is interesting and easy to follow, and presents a quite complete picture of this complex situation: starting with the demonstration of the excitation mechanism (optical spin transfer) to the dynamics of DWs excited by those mechanisms. **My only criticism is that some of the results could be more extensively discussed (as per some of the remarks below).**

I recommend the publication of this manuscript after the authors address my minor remarks below. My remarks and questions are roughly in order of appearance in the main text.

We thank the reviewer for evaluating our manuscript and noting the relevance of our work. We also appreciate his/her very precise and deliberate suggestions and comments, which gave us the opportunity to improve our manuscript and clarify some ambiguities. Below, we prepared a point-by-point response to the reviewer's comments together with the changes made to the manuscript.

1. The 'inertial displacement' of the DW by relaxation from a deformed state has been observed before, and these works should be cited: to my knowledge, it was first shown in J.-Y. Chauleau et al., Phys. Rev. B 82, 214414 (2010).

We thank the reviewer for pointing us to this very interesting paper describing DW motion after the pulse was applied due to the spin-transfer torque excitation induced deformation of a transverse wall into a vortex wall. We have now included the corresponding reference (now Ref. 8) as an example for inertial transient displacement after a ~ 1 ns pulse excitation.

2. page 4 first paragraph : What do you mean by "DW propagation is coherent"?

With coherent DW propagation we wanted to emphasis that the DW in our system propagates continuously and remains connected even if pinning acts locally only on part of the DW. In this case, partial DW pinning affects the entire wall along its entire extension. We understand that the wording "coherent" is misleading and have exchanged the word "coherent" with "uniform" on page 3 and 4. We also have added the sentence at the end of page 3: *In this case, the DW propagates continuously and remains connected so that local DW pinning affects the entire wall over its whole extension.*

3. page 4 last line: in " $K_E = 1200 \text{ Jm}^3$ " should be instead " $K_E = 1200 \text{ J/m}^3$ ".

We corrected the units of the effective perpendicular anisotropy constant, now at the beginning of page 5.

4. Perhaps state the width of the track in the legend of fig2E.

We added the track width ($4\mu\text{m}$) in the caption of Fig. 2E as the reviewer suggested and added also the device dimension in Fig. 2D.

5. page 5 2nd paragraph: When describing the "basic idea of [the] experiment" you state that the elastic force acts over the $1\mu\text{m}$ wide cross. However, you show measurements shown in crosses of different widths, and indeed the cross described in the paragraph just before is $4\mu\text{m}$ wide.

Indeed, the elastic bubble model is not limited to $1\mu\text{m}$ wide crosses but describes DW motion in all devices used in our experiments shown up to a $6\mu\text{m}$ track width (Fig. 2). We therefore modified the corresponding sentence at the end of page 5: "The basic idea of our experiment is to exploit the elastic restoring force which acts continuously throughout the entire cross (of a width up to $6\mu\text{m}$ in our study) against the expansion of the DW which is driven by individual ~ 100 fs LPs."

6. fig 3 (and fig S4): There is little discussion on the evolution of the oSTT effect with LP energy. For example, there's seems to be a saturation of the the effect with LP energy, as seen by the plateau of $(B_{DP+} - B_{DP-})$. Have you got an idea why? Have you perhaps reached the full DW distortion towards a Neel orientation? or perhaps it's due to heating?

We thank the reviewer for bringing up this important point. Increasing the LP energy leads to both enhancing the oSTT and increasing the sample temperature due to laser heating. We always initialise our experiments by field driving the DW to the cross entrance as shown schematically in Fig 3A. After sample saturation, we first nucleate a reverse domain, e.g. with a positive magnetisation orientation as shown in Fig 3A, before we apply a positive magnetic field B_A which must be smaller than the DW depinning field without LPs: $B_{DP} (\sim 0.4\text{mT}) > B_A > B_{PR} (0.1\text{mT})$.

We then identify the depinning field for left (right) helicity (σ^{\pm}) by applying B_A which, depending on its polarity, either assists depinning ($B_A > 0$) or counteracts ($B_A < 0$) DW depinning from the geometrical pinning site.

We found that LPs with σ^+ polarisation enable depinning without applying assisting B_A at LP energy densities above 12 mJ/cm^2 . At high enough LP energy, depinning is even possible at applied field counteracting the DW propagation. However, the magnitude of the negative fields we are able to apply without already depinning the DW from the cross entrance is limited by the propagation field $B_{DP} \sim -0.1\text{mT}$. We are therefore restricted to only small magnitudes of negative fields even at higher LP energy densities giving rise to the saturating behavior for LPs with σ^+ polarisation shown in the inset of Fig 3.

In the case of linearly polarised LPs (σ^0 polarisation), we do not expect oSTT contributions. However, we also observed a decrease of the depinning field with increasing LP power for σ^0 polarised LPs. We associate this to laser induced sample heating and the corresponding decrease of the geometrical pinning potential.

Note, that for linearly polarised light we do not observe DW depinning without assisting B_A up to the highest LP energy densities above 30 mJ/cm^2 used in our experiments. The observed saturation behaviour of $B_{DP}(\sigma^0)$ implicates saturation in sample temperature with increasing laser power. We assign this behaviour to a saturation in the generation of photo carriers at high LP energy densities above 20 mJ/cm^2 .

The saturation behaviour of B_{DP} observed for linearly polarised LPs (no oSTT) is also reflected in the saturation of $B_{DP}(\sigma^-)$ with increasing LP energy density. The oSTT at σ^- photon polarisation is acting against DW propagation towards the cross and indeed we observe $B_{DP}(\sigma^-) > B_{DP}(\sigma^0) > B_{DP}(\sigma^+)$ in all our measurements with a positive magnetisation orientation of the reversed domain. The opposite polarisation dependency is observed at negative magnetisation of the nucleated domain (Fig 4).

To describe the experiment in more detail we have changed and extended the text at page 5 and 6 as follows:

The measurements shown in Fig. 3 are performed at a 90K sample temperature. We obtain similar results when performing the measurements also at higher (95K) and at lower (75K) sample temperatures, as shown in the Supplementary Information. At these temperatures, LPs with a wavelength $\lambda = 750 \text{ nm}$ excite photo-electrons slightly above the bottom of the GaAs conduction band so that for a circularly polarised incident light, photo-electrons become spin-polarised with the degree of polarisation approaching the maximum theoretical value of 50%. To avoid the difficulty with aligning our $\sim 1 \mu\text{m}$ Gaussian spot on top of a $\sim 20 \text{ nm}$ wide DW, we employ the experimental procedure sketched in Fig 3.A. First, a straight DW is positioned at the cross entrance. Then, the LP spot is placed $10 \mu\text{m}$ away from the DW on the reversed domain side and a magnetic field B_A is applied with $B_{DP}^0 (\approx +0.4 \text{ mT}) > B_A > B_{PR} (\approx -0.1 \text{ mT})$. In this field range and without LP irradiation, the DW remains pinned at the cross entrance. The LP spot is then swept at a rate of $2 \mu\text{m/ms}$ for $20 \mu\text{m}$ along the bar so that the initial DW position is crossed by the spot and approximately $\sim 10'000$ ultra-short LPs time-separated by $\sim 10 \text{ ns}$ expose the DW. The lowest applied magnetic field at which the DW depins from the cross and which is lower than B_{DP}^0 due to the LPs corresponds to the depinning field B_{DP} . The dependencies of B_{DP} on the LP energy density for circularly polarised σ^+ , σ^- and linearly polarised σ^0 LPs are shown in Fig 3.B. First, we recognize a reduction of B_{DP} with increasing energy density for all three LP polarisations. In case of the linear polarisation, i.e., without oSTT contributions, we attribute the reduction of $B_{DP}(\sigma^0)$ only to the LP induced sample heating. Importantly, we do not observe DW depinning without applying $B_A > 0$ up to the highest LP energy densities used in our experiments of more than $\sim 30 \text{ mJ/cm}^2$. At large LP energy densities above $\sim 20 \text{ mJ/cm}^2$ we observe a saturation of $B_{DP}(\sigma^0)$ with increasing LP energy density implying

that LP heating does not increase anymore. We assign this behaviour to the saturation of photo carriers generated at very high LP energy densities.

For circularly polarised LPs, additional contributions from the oSTT are present. We observe for all measured LP energy densities that $B_{DP}(\sigma^+) < B_{DP}(\sigma^0) < B_{DP}(\sigma^-)$ for the positive magnetisation orientation of the nucleated domain. In case of σ^+ polarised LPs and at high enough LP energy densities (above $12\text{mJ}/\text{cm}^2$) the DW depins without an applied magnetic field (and even at small negative applied magnetic field which opposes DW expansion). For σ^- polarised LPs and the same initial domain configuration, we do not observe the zero-field DW depinning up to the highest LP energy density used in our experiments (see inset of Fig. 3B). Instead, we again observe saturation of $B_{DP}(\sigma^-)$ above $\sim 20\text{mJ}/\text{cm}^2$. We attribute the difference in the saturation values of $B_{DP}(\sigma^-)$ and $B_{DP}(\sigma^0)$ to the effect of the oSTT acting against depinning for σ^- polarised LPs.

7. page 7 para 2: you conclude that σ^- LPs drive the DW away from the cross but, at this paragraph, you have only shown the opposite effect. The results for pulling DWs with σ^- light is shown only later...

From Fig. 3, where $B_{DP}(\sigma^-) > B_{DP}(\sigma^0)$ for all observed LP energy densities we conclude already that σ^- polarised LPs indeed oppose DW depinning. That this is a result of backward DW motion away from the cross is an assumption, which is eventually verified by measurements shown in Fig. 4B. We changed the text as follows:

From the measurements shown in Fig. 3 we can conclude that for the given initial domain configuration, the oSTT generated by σ^+ (σ^-) polarised LPs assists (opposes) DW depinning. The measurements confirm that only σ^+ polarised LPs can move the DW beyond the maximum of the pinning barrier at the cross centre.

8. page 7 para 2: you say that "DW velocities of more $1\mu\text{m}/\text{ns}$ [$10^3\text{m}/\text{s}$] are unrealistically high". How so? can you estimate the Walker velocity? Is there another limitation? (heating? nucleation?)

From the simulation of the DW propagation within the LLB approach we obtain low time-averaged DW velocities for the oSTT driven motion. The velocity vs. excitation profile which we calculated using the LLB theory suggests that the oSTT driven DW motion has field-like character, (velocity of $\sim 5\text{m}/\text{s}$ reached at the WB drops down after the WB instability). Hence, the WB velocity for our oSTT DW motion is found to be much smaller than $\sim 1\text{km}/\text{s}$ required to explain DW depinning without inertia. Note that the WB-velocity in our case is of a similar magnitude to the WB velocities estimated for current and magnetic field driven DW motion in a similar GaMnAsP film (E. De Ranieri, et al., Nature Materials 12, 9, 808 (2013)).

Furthermore, the magnon group velocity of $\sim 200\text{m}/\text{s}$ in our GaMnAsP film is well below $1\text{km}/\text{s}$. We have added Fig. S4 in the Supplementary Information where we calculated the average DW velocity as a function of the LP energy density and added a short discussion on the magnon velocity limit in the revised supplementary Information chapter 2.2.

Figure S4: DW velocity time averaged during the 12.5ns dark phase after an oSTT pulse as a function of LP energy density calculated in the LLB approach

9. eq.s 1&2 (page 8-9) : you should write explicitly the coupling term between the two equations (the exchange field term in Heff created by s) in the main text, and not just in the additional materials. It is a central part of this model and critical for the understanding your model.

We followed this suggestion and added the expressions in the main text, below equation (1) and (2) on page 9.

10. page 9 para 4: from your text, one may think that the action of the photoelectrons on m happens after the recombination ("...so that a significant change of m [...] happens after the photo electrons recombined."). If I understood correctly, the photo electrons only distort the magnetisation of the DW until they recombine, and the DW only moves later (through relaxation) due to its slower dynamics (?). Perhaps you meant that the major effect of the photoelectrons (the DW propagation) are only evident well after they have recombined?

We thank the referee for this comment. His/her comment describes exactly how we understand the ultrashort circularly polarised LPs driven DW motion. For clarity, we add the sentence on page 11: "*The photo-electrons only distort the DW while its subsequent motion is driven by the relaxation of the DW towards its equilibrium profile.*"

11. page 10 (& addt mat): you state (and show in the additional materials) that a CW excitation is less power efficient in propagating the DW, something you have seen experimentally and in simulations. A bit more information in the main text as to why is this the case would be interesting.

DW motion driven by ultrashort pulses is very different from DW motion driven by cw-excitation at equal *time-averaged* LP energy density. In our experiment with ~ 100 fs pulses repeating every 12.5 ns, the pulse amplitude is 5 orders of magnitude larger than the steady-state cw-excitation amplitude.

Experimentally, we observe that pulse-driven DW motion driven by the self-propelling relaxation of the DW is much less affected by pinning than DW motion excited by continuous and lower energy-density light. This is due to the fast and non-uniform acceleration of the heavily deformed DW shortly after the pulse.

Our observation is confirmed by simulations where we consider only the geometrical DW pinning potential due to our cross structure. Indeed, pulse-driven DW motion overcomes the pinning potential at lower averaged laser power than the cw-excited domain wall. (Fig S7a)

We modified the corresponding paragraph: *"We finally remark that the helicity dependent DW motion can be also realised by a continuous light excitation. However, LP-excited DW motion driven by the self-propelling relaxation of the DW is less affected by pinning and, therefore, more efficient than DW motion excited by continuous and lower energy-density light. This is due to the fast initial acceleration of the heavily deformed DW shortly after the pulse."*

12. fig 1B: it seems there are some white semi-transparent rectangles over the white domains in the MOKE differential image. This may give the idea to the reader that the experimental image is less noisy than it actually is... I would suggest to replace them by empty shapes with borders, so as to still indicate the position of the structure while still showing your MOKE image.

We followed the suggestion of the referee and removed the white semi-transparent rectangles in Fig. 1B.

Reviewer #2 (Remarks to the Author):

The authors report on DW motion induced by circularly polarized light pulses in GaMnAsP bars. This work is a continuation of their previous work [ref 13] published last year. They investigated the mechanism of optical Spin Transfer Torque, where the photon momentum exerts a torque on the DW magnetization leading to DW motion. The novelty of the present work is the use of ultrashort (sub ps) pulses. That offers them access to DW dynamics at the picosecond time scale. In particular they claim to observed an inertial DW motion between the pulses.

This work belongs to the emerging subject of light induced magnetization manipulation, as magnetization reversal or DW motion. A growing number of publications have reported such effects, in both ferromagnets or ferrimagnets. Nevertheless it has emerged recently that heating related effects also play a major role in the magnetization dynamics. In their paper Janda et al., have addressed this issue. However as this point is critical and as the author's arguments to exclude it do not appear convincing enough, I cannot recommend this work for publication in Nature Comms.

We thank the reviewer for critically evaluating our manuscript and also noting the relevance of our work. In particular we appreciate his very important and critical point on LP heating and in particular helicity dependent heating which could, in principle, also cause helicity dependent DW motion. As we show below and in the revised Supplementary information, we have performed additional experiments which clearly evidence that we can exclude heating-dependent effects as the dominant driving mechanism of our observed helicity dependent DW motion.

My main concern is that all the experiments were performed very close to TC, which makes very hard to exclude any thermal effect as the main source of DW motion. In their previous PRL, the authors selected their substrate temperature to form easily domains. Here they benefited from a strip line for nucleation that could have given them access to lower temperatures. In this paper, unfortunately, no additional measurements were presented at lower temperatures. I would have expected that oSTT to be as efficient, given the very high quality of the sample, in particular very weak pinning energy. That would have given a stronger support to their claim.

We thank the reviewer for bringing up this point. Measurements at a lower than 90K temperature (75K) were shown already in the original Supplementary Material (Fig. S5). However, the data were wrongly labelled as 90K. In the revised manuscript we have corrected this typo. In addition we performed experiments at a higher temperature of 95K. The comparison of the 75K and 95K measurements, shown below, indeed confirms that oSTT is present independently of the sample temperature.

Figure S2 C,D: Polarisation dependent depinning field $B_{dp}(\sigma^+, \sigma^-)$ as a function of LP energy density at different sample temperatures, $T = 95\text{K}$ and $T = 75\text{K}$.

At lower temperatures, depinning without additional applied magnetic field is realised at even lower laser powers. This observation supports the expectation of the reviewer that the oSTT becomes more efficient at lower temperatures. We added Fig S2 C, D in the Supplementary information section of section 1.

To perform experiments at temperatures below 75K was not possible because we failed to initialise a single DW at the cross entrance at those low temperatures. During the initialisation procedure we first nucleate a reversed domain underneath the nucleation stripe and then move the DW by applying a small magnetic field to the cross entrance where it gets geometrically pinned. However, with decreasing temperature, the depinning of the DW from the nucleation area requires increasingly larger fields. If the field becomes too large, domain nucleation in other areas of the Hall bar device occurs and additional DWs approach the Hall cross making the realisation of the experiment impossible. Nevertheless, the measurements performed at 75K, 90K and 95K confirm the consistency of the oSTT picture.

At their experimental temperature, the authors did consider heating related effects before rejected them as the main source of DW motion. In particular they dismissed any magnetic circular dichroism related effects based on the fact that they observed DW motion over a spectral range for the pump laser that does not match the MCD one in GaMnAs. I cannot agree with this statement. I could find in the literature several papers that show dichroism curves reproducing the trend in fig S4.b:

For perpendicular magnetization samples, Sun et al. PRB 83, 125206 and Riahi et al. JMMM395,340 measured Kerr angle and Kerr angle + ellipticity respectively vs wavelength. Although Kerr is not MCD, these quantity are highly connected. And they do observe a reduction of the Kerr angle and ellipticity at shorter wavelengths. In another study Dobrowolska et al. Nat. Mat. 11, 444 worked in planar samples with at high perpendicular magnetic fields. They observe a similar trend.

They also rejected this MCD effect based on the fact that most of the light is absorbed into the substrate and therefore the MCD related heating should be weak. However I went through these two publications, Khorsand et al. PRL 108, 127205 and Gorchon et al. arXiv <http://arxiv.org/pdf/1604.06441v1.pdf>. They both assessed the role of MCD in dichroic materials. The latter publication

spotted clearly that even a small MCD (0.5 %) could lead to helicity dependent optical magnetic switching driven by heat. In particular they indicate that this effect should be stronger in "materials with large MCD, small dipolar fields, limited in-plane diffusion and strong temperature of the anisotropy close to TC", matching very closely the properties of GaMnAsP. Although they discussed the case of magnetic switching, I believe that a proper analysis of DW motion should be undertaken in the light of this mechanism. In particular, the effect of the laser pulse ellipticity remains tenuous and increase only when approaching TC. The missing temperature dependence experiment does not help to distinguish between heat and oSTT related effects.

We agree with the referee that it is useful to present additional supporting evidence that excludes MCD as the driving mechanism for the observed helicity dependent DW motion. We have, therefore, performed new experiments (described below and in the Supplementary information, section 3.5: "MCD induced temperature gradients") which allow us to identify the sign and estimate the magnitude of the temperature gradient generated by the MCD between two oppositely magnetised domains. From these experiments we found out that the MCD-generated heat-gradient would induce DW motion in the opposite direction than observed in our helicity-dependent experiments. Therefore, MCD may affect but cannot explain our experimental observations.

In these new experiments, we evaluate the temperature variation generated by the MCD under exactly the same experimental conditions (same optical setup, applied laser powers, spot size, substrate temperature, sample mounting in the cryostat, etc.) as used in the DW motion experiments described in the main text and the Supplementary material. We use a 6 μ m wide, 18 μ m long (contact-to-contact) bar-device patterned from the GaMnAsP/GaAs film which was also used for the DW experiments. LPs are focused at the centre of our bar as shown in Fig S9 of the revised Supplementary material.

Figure S9: Cross bar device for 4p measurements: Left: Laser spot focused to the centre of the bar between the two cross-contacts; Middle, (Right): MOKE micrographs at positive, (negative) magnetisation orientations.

We evaluate the temperature variation at opposite saturation magnetisations from the resistance variation detected in our sample at a fixed LP polarisation. We employ a sensitive double Lock-in technique and compare magnetisation dependent resistance variation with the temperature dependence of the GaMnAsP film resistivity.

To estimate the temperature variation at the irradiated spot position from the resistance variation measured in 4-point geometry over the whole bar device, we employ a simple resistor network model described in Fig. 4. The 18 μ m long and

6 μm wide bar is divided into 27 (2 μm \times 2 μm) squares. Only the central square is irradiated having the resistance R_{sq}^L . All other squares remain in darkness having the equal resistances of R_{sq}^D .

Within this simple resistor network model, we can estimate the temperature variations generated by the MCD at the irradiated square of the device by measuring the total device resistance variation at opposite saturation magnetisations when comparing the measured data to a reference temperature measurement $\Delta(T) = R(T)/R(90\text{K})$, Fig. S10, right. Since only the irradiated spot can be compared with this reference measurement, we need to relate the sample temperature variation of the spot to the measurable resistance variation of the total device as shown in Fig S10 (left). In the frame of this approximation, the ratio of irradiated and non-irradiated square resistances is estimated to $\Delta = R_{sq}^L / R_{sq}^D = 27 R_T^L / R_T^D - 26$; (R_T^L and R_T^D are the measurable total bar device resistances with and without spot irradiation).

To avoid any small alternations of our focused LP spot in intensity, spot position, etc., we measure the resistance variation due to the MCD by changing periodically the sample saturation magnetisation (at a frequency of 0.2 Hz) and keeping simultaneously the LP polarisation fixed. A first Lock-in amplifier measures the 4-point resistance as the response to an alternating probe current ($f = 123\text{Hz}$). After the subtraction of an offset, the output of the first Lock-in is amplified by a factor $\times 100$. The resulting signal feeds into a second Lock-in, which amplifies the signal with reference to the alternating saturation magnetisation. As a result, we obtain $R_{T,+Mz}^L - R_{T,-Mz}^L$.

spot (2x2 μm^2), sample: 3 parallel resistors of 2 μm width, 18 μm length

Figure S10: Model to estimate temperature variations due to MCD: Left: Simple resistor network model; Right: Resistance variations as a function of the sample temperature.

We evaluate $dT = T(R_{sq,+Mz}^L) - T(R_{sq,-Mz}^L) = \alpha(\Delta_{+Mz} - \Delta_{-Mz})$ with $T = \alpha \Delta$ and α is obtained from the linear slope of the temperature dependence of the sample resistance around $T = 90\text{K}$. Since $\Delta_{+Mz} - \Delta_{-Mz} = 27(R_{T,+Mz}^L - R_{T,-Mz}^L) / R_T^D$, we can estimate $dT = 27\alpha(R_{T,+Mz}^L - R_{T,-Mz}^L) / R_T^D$ from our measured data.

In Fig S11 (a,b) we show the LP polarisation dependent temperature variation at the irradiated spot between positive and negative saturation magnetisations. At $T = 90\text{K}$ substrate temperature, the temperature variation is of the order of $+(-)$ 200mK for circularly polarised $\sigma+(\sigma-)$ LPs. Hence, the temperature of the $\sigma+(\sigma-)$

LPs irradiated film with positive (negative) magnetisation becomes higher than the film with negative (positive) magnetisation orientation.

Figure S11: (a): Temperature variation measured at $T = 90$ K substrate temperature and for a laser power of $10\text{mJ}/\text{cm}^2$ as a function of LP polarisation. The polarisation is varied by a $\lambda/4$ - waveplate. A constant offset is subtracted from all data points. Before taking a data point, we always realigned the focused laser spot to the bar centre. The discrepancy between the measured angular dependence of the temperature variation from the expected $\cos(2\phi)$ behaviour is due to unintentional polarisation effects in some of the optical components in our setup. σ^+ and σ^- polarised LPs correspond to $\phi = 135$ deg and $\phi = 195$ deg, respectively. As expected for MCD related heating, we find largest temperature variations for $\phi = 135$ deg (red line) and $\phi = 195$ deg (blue line) polarisations.

(b) MCD induced differences in temperature variation between σ^+ and σ^- as a function of LP energy density for 3 different substrate temperatures, 75 K (blue), 90 K (green), and 117 K (red). Note that the MCD induced heating disappears above the Curie temperature $T_c = 115$ K.

(c) Static DW profiles obtained by micromagnetic simulation based on the experimentally obtained values at $T = 90$ K for out-of-plane anisotropy $K_\perp = 1500$ Jm $^{-3}$ and saturation magnetisation $M_s = 18.2$ kA/m. We use $A = 50$ fJ/m for the exchange stiffness parameter.

In our oSTT experiments shown in Fig 3 of the main text, we saturate the magnetic bar in a strong negative magnetic field and nucleate a reversed domain at the left side of the bar with positive magnetisation orientation. When irradiated with σ^+ (σ^-) polarised LPs, MCD heats up the nucleated magnetic domain more (less) than the rest of the magnetic film with negative magnetisation orientation. In the experiment, we observe DW depinning and motion towards the area with negative magnetisation only when the DW is irradiated with σ^+ polarised LPs. If we repeat the experiment with inverted magnetisation (positive saturation, nucleated domain with negative magnetisation orientation) the DW moves towards the positive magnetisation orientation only when irradiated with σ^- polarised LPs.

Hence, we always observe that the circularly polarised LP exposed DW moves towards the colder region, which excludes the MCD origin of the observed DW motion.

Considering a DW width of ~ 50 nm (Fig 5c), we estimate a MCD generated heat gradient of $\sim 6 \times 10^6$ K/m. This value is smaller than the heat-gradient generated by the focused light spot independent of polarisation.

We present the discussion regarding the MCD in chapter 3.5 of the supplementary information.

Another argument put forward by Janda et al. is the inertial motion observed after the light pulse. Here again, I am a bit concerned as heat dissipation in the magnetic layer and substrate occurs on a time scale far longer than the sub picosecond pulse and the repetition time. That means that right after the pulse, the anisotropy, the pinning energy and the magnetization remain very weak which can lead to very different dynamics under their applied magnetic field, especially high DW velocities.

We have taken into account heat-related reductions of the anisotropy, saturation magnetisation etc by using the LLB approach. From our simulation we obtain a maximum velocity at the WB instability of only ~ 5 m/s. However, the limiting velocity for any texture to propagate in a magnetic film is the magnon velocity, which we estimate to be approx. 200 m/s in our GaMnAsP film at 90K. Even the magnon velocity is too low to enable depinning of the wall without inertial motion. Additional heating of the film will further reduce the limiting magnon velocity. We add a short discussion about the velocity of our DW in the revised Supplementary information 2.2.

I was surprised by the negative points in the depinning field difference observed at low power and at 880 and 750 nm in Fig S4A lower plots. Could the authors comment on that?

The error-bars of the two points derived from the depinning fields for $\sigma^{+/-}$ measured at very low LP energy density of ~ 2.5 mJ/cm² are too large to be able to interpret them as non-zero data-points.

I found the introduction and justification of this work about the interest of inertial and massless DW motion rather confusing (a massless DW is the best but probably also inertial...).

We changed the introduction on page 3 to clarify the relevance of inertial DW motion for ultrafast excitation:

"The aim of our study is the demonstration of a micrometer-scale DW displacement by circularly-polarized, ultra-short laser pulses (LPs). Our experiments are in the regime where the external force generated by the LP acts on the picosecond time-scale over which the expected sub-nanometer DW displacement would be orders of magnitude smaller than the DW width and insufficient for any practical DW device implementation. Inertia allowing for a free transient DW motion is the key here that enables the operation of the DW devices in the regime of the ultra-short optical excitations, rather than being a factor limiting the operation of the opto-spintronic DW devices."

Reviewer #3 (Remarks to the Author):

In the article "Inertial displacement of a domain wall excited by ultra-short circularly polarized laser pulses", T. Janda et al. describe domain wall motion driven by femtosecond circularly polarized pulse. The driving force was attributed to the optical spin transfer torque. The authors claims that the motion is inertial displacement.

This work is original and interesting, and also important for application. The physical interpretation of the experimental data is mostly convincing. The references are appropriate. The manuscript is well written.

Their experimental finding will stimulate a new trend in the research community. Several future research questions arise; For example, how fast can the domain wall move with this inertial mechanism? How is the inertial mass of the motion defined? What limits the length of the inertial displacement by a laser pulse?

We thank the reviewer for his/her positive comments noting that he/she finds our work original and interesting and relevant for applications. We also thank the reviewer for his additional remarks and questions which we hope to answer in an adequate manner. Before answering the specific questions we also would like to comment on the reviewer 3 introductory remarks:

I1) How fast can the domain wall move with this inertial mechanism?

It is established that the magnon group velocity gives the limiting speed for any texture to propagate in a given magnetic material, which in our case is approx. 200 m/s. From our LLB simulations we observe DW velocities of only up to 5m/s. If we increase the oSTT excitation further, we overcome the so-called Walker Breakdown (WB) instability and the DW velocity decreases again. Stabilising the DW structure, e.g., by DMI or by introducing additional anisotropy via mechanical strain, can shift the WB to higher values and allow for achieving higher DW velocities.

Fig. S4: DW velocity, time averaged during the 12.5ns dark phase after a oSTT pulse, as a function of laser power for GaMnAsP film without (red) and with uniaxial inplane anisotropy K_u^y (black). The uniaxial inplane anisotropy can be introduced by mechanical strain (E. De Ranieri, et al., Nature Materials 12, 9, 808 (2013)).

I2) How is the inertial mass of the motion defined?

We envision the oSTT driven DW dynamics as an exchange between “internal” (or “potential”) and “kinetic” energy. By internal energy we mean the energy that the DW stores in its internal structure. This internal energy increases if the DW distorts its profile by external forces, e.g. by the action of the oSTT. From our simulations we found that the DW indeed strongly deforms its profile during the action of the oSTT so that the DW has increased its internal energy after excitation. After the oSTT pulse, the DW starts a “self-propelled” propagation by transforming part of this stored internal energy into motion (kinetic energy), magnons emission and heat.

Our simulations provide the solution of the full spatially dependent LLB equations. DW mass appears explicitly only in an effective model that reduces the complexity of the full LLB equations by treating the DW as a point particle and obtaining the expression for its momentum from the equations of motion. For example, in a 1D model of the DW one obtains a DW mass that is inverse proportional the DW width and the anisotropy energy [E. Schloemann, J. Appl. Phys. 43, 3834 (1972)]. We therefore think that inertial mass is not an instructive concept for our ultra-short LP driven DW motion where the DW deformation is changing during its inertial motion.

I3) What limits the length of the inertial displacement by a laser pulse?

The DW experiences ‘friction’ (due to Gilbert damping and magnon emission). It is therefore losing internal energy when it is moving. And it can move until it has relaxed to its unperturbed static configuration. Therefore, the length of the inertial displacement depends on the degree of deformation, the strength of the Gilbert damping, and the degree of spin waves emission.

The remaining comments of the reviewer are addressed below:

1) The laser heating also lead to inertial displacement of a domain wall. What is the mechanism?

That heat gradients in the magnetic film can cause domain wall motion towards the hot part of the magnetic film, was shown earlier by J.-P. Tetienne, et al., Science 344, 1366 (2014) and others. From the thermodynamic point of view the mechanism can be understood by temperature dependencies of material parameters such as saturation magnetisation, magnetocrystalline anisotropy and thus the effective anisotropy and exchange stiffness all vary with temperature resulting in a spatial variation of all these material parameters, in particular of the domain wall energy. At higher temperature, the domain wall energy is lower and the domain wall will move towards the hotter region as it lowers its energy. A more microscopic description considers that at the hot end, magnons are thermally populated. Diffusive magnon flow from the hot to the cold end takes place together with a transfer of angular momentum to the domain wall causing it to move in the opposite direction to the magnon-current, i.e., the domain wall moves also towards the hot area.

In our case, heat-gradient driven domain wall motion can be identified but we exclude it as the mechanism behind the helicity dependency of the DW motion. Supporting experiments performed in response to the comments of Reviewer #2

and included in the revised Supplementary information confirmed that the helicity dependent heating is acting against the observed DW motion.

2) In inset in Fig. 3B, the data for linearly polarized LP should be added. The additional data will provide a strong evidence for oSTT mechanism in the case of circularly polarized LP.

We changed Fig. 3B and added the missing data for linearly polarised LPs which are in agreement with the other datasets for left and right circularly polarised LPs.

Reviewer #1 (Remarks to the Author):

I am satisfied with the response of the authors to the points I raised in the first review, and I recommend the publication of this manuscript.

Reviewer #2 (Remarks to the Author):

The authors have answered in details to my criticisms, including new and extensive experimental and simulation results. I am now inclined to revise my judgment and recommend its publication in Nature Comm.

Just a brief remark about the author answer, I did not expect the oSTT efficiency to be higher when reducing the temperature, but rather to be similar. This improved efficiency is actually a bit surprising to me, as domain wall velocity, in standard STT, increases as the temperature increases (as M decreases).

Reviewer #3 (Remarks to the Author):

I feel that the points raised in the first round of review have been satisfactorily addressed. I have no more comments and recommend the paper for publication in Nature Communications.